# Comparison of Antigen Retrieval Methods for Immunohistochemical Analysis of Cartilage Matrix Glycoproteins Using Cartilage Intermediate Layer Protein 2 (CILP-2) as an Example

**DOI:** 10.3390/mps7050067

**Published:** 2024-08-24

**Authors:** Taavi Torga, Siim Suutre, Kalle Kisand, Marina Aunapuu, Andres Arend

**Affiliations:** 1Department of Anatomy, University of Tartu, Ravila 19, 50411 Tartu, Estonia; siim.suutre@ut.ee (S.S.); marina.aunapuu@ut.ee (M.A.); andres.arend@ut.ee (A.A.); 2Department of Internal Medicine, University of Tartu, L. Puusepa 8, 50406 Tartu, Estonia; kalle.kisand@ut.ee

**Keywords:** antigen retrieval, cartilage, CILP-2, immunohistochemistry, osteoarthritis

## Abstract

The aim of this study was to compare different antigen retrieval methods to improve the outcome of immunohistochemistry (IHC) performed on osteoarthritic (OA) cartilage obtained from total knee replacement operation. A voluminous and dense extracellular matrix of articular cartilage inhibits antibody penetration, and therefore, proteins present at low concentrations and masked during fixation may need antigen retrieval to enhance an IHC outcome. We focused on the IHC detection of a minor but diagnostically promising cartilage glycoprotein, CILP-2 (cartilage intermediate layer protein 2), to demonstrate the effect of four different protocols: (1) heat-induced epitope retrieval (HIER), (2) proteolytic-induced epitope retrieval applying proteinase K and hyaluronidase (PIER), (3) HIER combined with PIER, and (4) no antigen retrieval (control). A semi-quantitative staining assessment based on the CILP-2 staining extent was applied. Out of the tested antigen retrieval protocols, the best CILP-2 IHC staining results were achieved by PIER. Combining PIER with HIER did not improve CILP-2 staining in the given experimental setting. Rather the opposite, the application of heat reduced the positive effect of PIER on CILP-2 staining and resulted in the frequent detachment of sections from the slides. Our findings emphasize the need for proper adaptation of antigen retrieval protocols for IHC to maximize the quantitative evaluation of minor matrix proteins in OA articular cartilage samples.

## 1. Introduction

The cartilage matrix consists up to 15% of glycoproteins, and extensive research has been performed to describe their role in maintaining the functional structure of the articular cartilage and their involvement in different degenerative joint diseases [1]. Changes in the cartilage matrix content of glycoproteins and different aspects of their biological functions have been associated with the loss of joint function and concomitant joint pain [2]. The cartilage intermediate layer protein (CILP) is one of many glycoproteins that have been in the focus of research of joint diseases, such as osteoarthritis (OA). CILP has been described to be localized in the intermediate zone of normal articular cartilage and downregulated in experimental OA [3]. To date, two CILP isoforms, CILP-1 and CILP-2, have been described. These isoforms have been demonstrated to be expressed in cartilage but also in extra-skeletal tissues [4]. The two CILP isoforms are quite homologous with about 50% of the structure being similar [5]. Nevertheless, the isoforms have been differently associated with the progression of cartilage damage during OA, although the literature contains many contradictions. For example, CILP-1 and CILP-2 display different expression patterns in cartilage in animal models of OA, with CILP-1 being upregulated and CILP-2 downregulated [3]. In contrast, a proteomic analysis of cartilage samples of hip OA patients showed increased levels of CILP-2 [6]. Since CILP-2 rather than CILP-1 has been suggested to be related to the progression of OA, it has been attributed some diagnostic value [3,7]. The structure of CILP-2 is a polypeptide of 1156 amino acids with a molecular mass of 126.291 kDa [8]. Similarly to CILP-1, CILP-2 is considered to comprise two distinct polypeptides with glycosylated sites and disulfide bonds. It should be mentioned that fewer glycosylation sites have been described in CILP-2 than in CILP-1 [8], which may contribute to the lower stability of CILP-2, as maintaining the conformation means relying more on its non-covalent bonds, such electrostatic and charge–charge interactions, hydrogen bonds, and Van der Waals interactions. Glycosylation is a way to improve the physical protein stability and reduce the effect of precipitation, pH, chemical and thermal denaturation, and aggregation [9]. It is therefore possible that the level of glycosylation has its own effect on epitope integrity and, thus, on the successful antigen–antibody reaction necessary for effective immunohistochemical (IHC) analysis.

When working with IHC methods, great attention must be paid to how the protocol variations can affect the outcome and interpretation of the results. Although many issues can be encountered during the validation of the suitable IHC protocol, we decided to focus on the epitope retrieval methods, because this affects the most essential step of the procedure—antibody binding to its epitope [10]. The retrieval is usually required, as epitopes can be masked during formalin fixation due to the cross-linking of amino groups on adjacent molecules [11]. As CILP-2 is found in the matrix of articular cartilage rich in collagen II (COL2) molecules containing amino groups [12,13], the possible masking of CILP epitopes could be an issue and a suitable method for antigen retrieval is often needed.

The most common epitope retrieval method, presented in the mid-1970s, was the proteolytic-induced epitope retrieval (PIER) method. Trypsin, Proteinase K, pronase, ficin, and pepsin are the most frequently used enzymes in PIER. The main objective of PIER is to degrade protein crosslinks by protein digestion. The effect of PIER depends on different factors: the concentration and type of the enzyme, incubation parameters (time, temperature and pH), and the time length of fixation [14].

Later, in the 1990s, the heat-induced antigen retrieval method (HIER) was also introduced. HIER is a regularly used pre-treatment procedure in IHC to modify the molecular conformation of “target” proteins of slide-mounted specimens by heated buffer solution. The common problem of HIER is the potential destruction of the antigenicity of the epitopes [10]. The heat resistance of proteins has been suggested to be related to their solubility, and most proteins are indeed considered heat-labile, which means that proteins are irreversibly denatured and precipitate upon heat treatment [15]. Heat resistance can also depend on the glycosylation of the protein [9]; therefore, caution is particularly advised to optimize a suitable IHC protocol for the evaluation of cartilage proteins, including CILPs.

In our study, we compared different combinations of HIER and PIER on the outcome of CILP-2 staining and found enzymatic retrieval to produce the most abundant staining in the given protocol settings.

## 2. Materials and Methods

### Sampling

Cartilage samples were obtained from four patients (two men and two women) undergoing primary unilateral total knee replacement operation (TKR) due to end-stage knee OA at the Traumatology and Orthopaedics Clinic of the Tartu University Hospital (Tartu, Estonia). The patients ranged from 58 to 65 years old. Written informed consent was received from all the patients before participation. This research project was approved by the Ethics Review Committee on the Human Research of the University of Tartu, and it is in accordance with the Declaration of Helsinki (1975). Cartilage biopsies were obtained from load-bearing sites of the tibial plateau of three patients. In one patient, three samples from different locations were taken to ensure the sufficient amount of material for testing and to check the site specificity of different staining protocols. A total of six samples were fixed in 10% buffered formalin solution for no more than three weeks and were decalcified in 24 h by SAKURA TDE^TM^ 30 fixing decalcification system (Sakura Finetek Europe B.V, Alphen aan den Rijn, The Netherlands). After that, the samples were embedded in paraffin with a vacuum infiltration processor (Tissue-Tek^®^ 5 Jr, Sakura Finetek Japan, Tokyo, Japan), and 4 µm thick sections of paraffin blocks were cut with microtome Ergostar HM 200 (Ergostar HM 200, Microm GmbH, Walldorf, Germany) and mounted on TOMO^®^ Adhesion Microscope Slides (Matsunami Glass, Kishivada, Japan) for IHC staining.

## 3. Antigen Retrieval Methods and IHC Staining of CILP-2

### 3.1. Antigen Retrieval Protocols

Four different protocols were compared from the aspect of antigen retrieval of 4 µm thick tissue sections: (1) heat-induced epitope retrieval (HIER) without proteolytic-induced epitope retrieval (HIER group), (2) proteolytic-induced epitope retrieval (PIER) without heat-induced epitope retrieval (PIER group), (3) HIER combined with PIER (samples treated with this antigen retrieval method formed the group designated as HIER/PIER group) and (4) no retrieval at all (control group). Heat retrieval was performed at 95 °C for 10 min using specific heat retrieval (decloaking) solution (Reveal Decloaker, Biocare Medical, Pacheco, CA, USA). Proteolytic retrieval was performed with 30 µg/mL Proteinase K solution (Merck KGaA, Darmstadt, Germany) in 50 mM Tris/HCl, 5 mM CaCl_2_ solution (pH 6.0) in 90 min at 37 °C. Thereafter, the same sections were treated with 0.4% bovine hyaluronidase in HEPES-buffered medium (Fertipro NV, Beernem, Belgium) for 3 h at 37 °C.

### 3.2. Description of the IHC Method for Detection of CILP-2

The cartilage sections on microscope slides were deparaffinized in xylene and rehydrated in ethanol solutions. Sections in groups PIER, HIER, and HIER/PIER received antigen retrieval protocols, as described in the previous paragraph, while the control group sections were kept in distilled water (summarized in Figure 1). Thereafter, the sections of all groups were treated similarly. To inactivate endogenous peroxidase, all sections were treated with 0.6% H_2_O_2_ (Honeywell Fluka^TM^, Seelze, Germany) for 15 min. Then, the sections were washed for 3 × 5 min in 1 × GibcoTM Phosphate-Buffered Saline (PBS, pH 7.4) solution (Thermo Fisher Scientific, Landsmeer, The Netherlands). To block non-specific binding, the sections were treated with Dako REAL Antibody Diluent (S2022, Dako Denmark A/S, Glostrup, Denmark). Thereafter, the sections were incubated with rabbit polyclonal antibody to CILP-2 (Atlas Antibody, Sweden, anti CILP-2 cat no HPA041847, dilution 1:100 in Dako REAL Antibody Diluent) overnight at +4 °C. The visualization of the primary antibody was performed using the commercial kit “Dako REAL^TM^ EnVision^TM^ Detection System, Peroxidase/DAB+, Rabbit/Mouse” (cat no K5007) (Dako Denmark A/S, Glostrup, Denmark)—the sections were incubated with secondary antibody for 1 h and after that with DAB for 10 min. Washing 3 × 5 min between each step after primary antibody incubation was performed in phosphate-buffered saline (PBS, pH 7.4) containing 0.07% Tween 20 (BioTop, Naxo, Tartu, Estonia). In negative controls of staining, the primary antibody was omitted. Finally, to counterstain the cartilage, samples were stained with Toluidine blue solution for one minute at room temperature and rinsed in distilled water. After that, the sections were dehydrated with ethanol and xylene and mounted with Eukitt^®^ medium (Sigma-Aldrich GmbH, Steinheim, Germany) before applying the coverslips. 

### 3.3. Formation of Study Groups and Assessment of CILP-2 Staining

Based on the four different antigen retrieval protocols, four study groups were formed: (1) HIER group, (2) PIER group, (3) HIER/PIER group, and (4) control group (described above in more detail). In each group, six articular cartilage samples were treated with the antigen retrieval method followed by IHC detection of CILP-2, as described above. Thereafter, semi-quantitative staining assessment was applied where scoring was based on the CILP-2 staining extent. The staining scores ranged from 1 to 5 (“1”: no staining; “2”: 1–25% of the slide stained; “3”: 26–50% stained; “4”: 51–75% stained; “5”: 76–100% stained) [16]. The score evaluation was performed under microscope at 400 times magnification by two independent observers in a blinded fashion; in the case of the six discrepancies, the specimens were reviewed together to formulate the final assessment.

### 3.4. Statistical Analysis

As the values were assumed not to be sampled from Gaussian distribution, evaluations of CILP-2 staining scores of study groups were allocated to nonparametric statistical analysis by applying the Mann–Whitney U-test. A statistical analysis was performed using the GraphPad Prism software platform (GraphPad Prism 10.2.3, San Diego, CA, USA). The level of significance was set at *p* < 0.05.

## 4. Results

The four different antigen retrieval protocols tested gave clearly different results, as can be seen from CILP-2 IHC scores of four test groups (*n* = 6) with various combinations of antigen retrieval methods (Figure 2). 

The highest staining scores were recorded in the PIER group, where abundantly rich IHC staining of CILP-2 was seen all over the slide (Figure 3C). In terms of CILP-2 staining grades, the PIER group was followed by the HIER/PIER group with moderate staining of CILP-2 (Figure 2 and Figure 3A,C, respectively). In the remaining two groups, HIER and control, CILP-2 staining grades were similarly low (Figure 2). Immunostaining was visible only in the superficial layer in the HIER group and in the superficial and middle layers in the control group (Figure 3B,D).

CILP-2 staining grades of these two groups with no proteinase application in the epitope retrieval were significantly lower as compared to PIER group (*p* = 0.004 vs. control group and *p* = 0.012 vs. HIER group; Mann–Whitney U-test, Figure 2). No statistical differences were noted between groups HIER/PIER and HIER (*p* = 0.069) and groups HIER/PIER and control (*p* = 0.084). Thus, in this study’s settings, heat induction did not improve antigen availability as the results of CILP-2 immunostaining scores were similar in the control and HIER groups. Furthermore, it has to be emphasized that heat retrieval with the application of high temperature caused the frequent loosening of sections, which was not the case with proteinase retrieval. 

An additional morphological finding was the distinctive appearance of CILP-2 staining in the territorial matrix around the isogenous groups of chondrocytes in the PIER and HIER/PIER groups (see Figure 3A,C and Figure 4). Such staining was absent in HIER-treated and control sections.

No specific staining was seen in negative controls where the primary antibody was omitted.

## 5. Discussion

In this study, we investigated how different antigen retrieval methods influence the detection of the protein of interest and the interpretation of the IHC results in human OA cartilage. Our specific focus was on glycoprotein CILP-2, which is thought to be involved in OA pathogenesis [3] and is therefore a valid subject of IHC research. If a tissue sample is fixed in formalin, as is the case with most of the routinely collected pathoanatomical specimens, the cross-linking of protein amino acid residues occurs, involving formaldehyde-derived methylene bridges, which block the antibody binding and therefore mask the antigen availability for IHC [17,18]. There are essentially two methods to retrieve the epitope in the tissue so that it can bind to the antibody. One possibility is to break the methylene bridges by heating, applying a method designated as heat-induced epitope retrieval (HIER), and the other option is the digestion of structures surrounding the epitope with the help of enzymes—a method called proteolytic-induced epitope retrieval (PIER) [19]. It is usually expected that HIER would give better results, as there is a risk that PIER may fail to unmask the epitope or destroy the tissue morphology and the antigen of interest, but it has also been suggested that a combination of HIER and PIER would give the best results [20,21]. An added effect of HIER together with PIER was also our working hypothesis when we started analyzing the IHC results in the articular cartilage. However, as we soon discovered, the optimal results were achieved with the enzymatic treatment. We found no beneficial retrieval effect of heating compared to non-treated articular cartilage and, furthermore, when heating was added to enzymatic retrieval, the final staining was significantly reduced compared to the sole use of proteinase K/hyaluronidase treatment. Moreover, heating resulted in the frequent detachment of sections from the slides.

We also found a distinctive appearance of CILP-2 staining in the territorial matrix around the isogenous groups of chondrocytes in groups that received the proteolytic treatment. This was especially distinctive in the HIER/PIER group, as the rest of the cartilage matrix seemed to lose the staining, while it was still preserved around the isogenous groups. This was noted in superficial, intermediate, and deep layers of articular cartilage, and it could be hypothesized that CILP-2 molecules in closer proximity of the chondrocytes were synthesized relatively recently compared to the rest of the tissue matrix and were therefore perhaps more resistant to the heat treatment.

It is complicated to precisely explain our results. Cartilage is a unique tissue characterized by extensive dense extracellular matrix, which inhibits antibody penetration. This makes it really important to choose an appropriate retrieval method for immunohistochemical research of the cartilage. Compared to other tissues where the application of HIER or a combination of HIER and PIER is optimal, in cartilage, the use of proteinases for antigen retrieval may be more relevant. This is supported by studies on skeletal tissues where PIER yielded better results than HIER [19,22]. Furthermore, PIER seems particularly useful as it does not seem to cause as extensive detachment of cartilage sections from glass slides as HIER. The poor adhesion of cartilage sections to microscope slides has been a challenge with HIER, as reported by others working on skeletal tissues [23].

It should also be noted that osteochondral tissues usually require proper decalcification. To protect the antigens of interest throughout the extended decalcification step, the skeletal tissues need to be thoroughly fixed (usually no less than 24 to 48 h), which again emphasizes the necessity of selecting effective antigen retrieval methods for skeletal tissues once the IHC has to be performed.

Yet another factor in determining the optimal antigen retrieval method and therefore the successful binding of the antibody to its antigen is the nature of the antigen molecule itself. The temperatures involved in the HIER protocols can be rather extreme, and the factors determining the thermostability of a protein (such as glycosylation) could have a significant effect on the outcome of IHC.

To address the limitations of current study, we have to acknowledge that we cannot rule out the possibility that OA-induced changes in the articular cartilage may impact the availability of epitopes as well as the outcome of the antigen retrieval. Since it has been shown that glycosylation increases the stability of a protein [9], the structure of the glycoproteins in OA-affected cartilage need not be the same as in metabolically and functionally normal tissue. In order to reduce the effect of the advancement of the OA on our results, sections from all the samples were treated with all the three retrieval protocols and a control protocol (no retrieval) to have a comparison within the sample. A dedicated chemical characterization of the heat-sensitivity of the molecules in pathological tissues should be performed to further prove our hypothesis. Also, while this study focuses on OA cartilage, the findings may not be applicable to other types of tissues or conditions. However, it is important to note that the immunohistochemistry of cartilage possesses unique characteristics that distinguish it from other tissues, and it may be necessary to apply distinctly cartilage-specific IHC protocols.

## 6. Conclusions

In conclusion, the three antigen retrieval protocols tested using prior CILP-2 IHC staining demonstrated that the best staining results were achieved by applying PIER, which combined proteinase K and hyaluronidase. Heat-induced antigen retrieval methods did not improve CILP-2 staining in the given experimental settings. Rather the opposite, the application of heat combined with proteolytic epitope retrieval reduced the positive effect of PIER on CILP-2 staining and resulted in the frequent detachment of sections from the slides. Our findings emphasize the need for the proper adaptation of antigen retrieval protocols for skeletal tissues and, thorough characterization of the nature of the epitopes and marketed antibodies, to maximize the outcome of IHC detections. This is especially relevant in OA-affected articular cartilage samples challenged by the dense structure of the extracellular matrix, which is in the process of constant pathological changes during the disease course.

## Figures and Tables

**Figure 1 mps-07-00067-f001:**
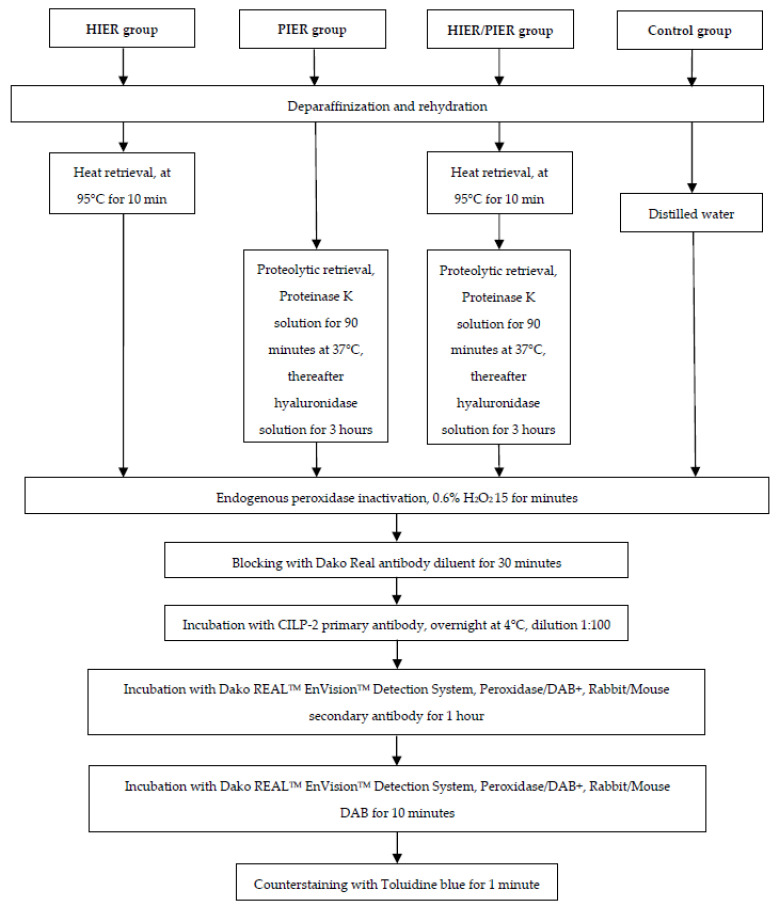
Comparison diagram of different antigen methods used in this study.

**Figure 2 mps-07-00067-f002:**
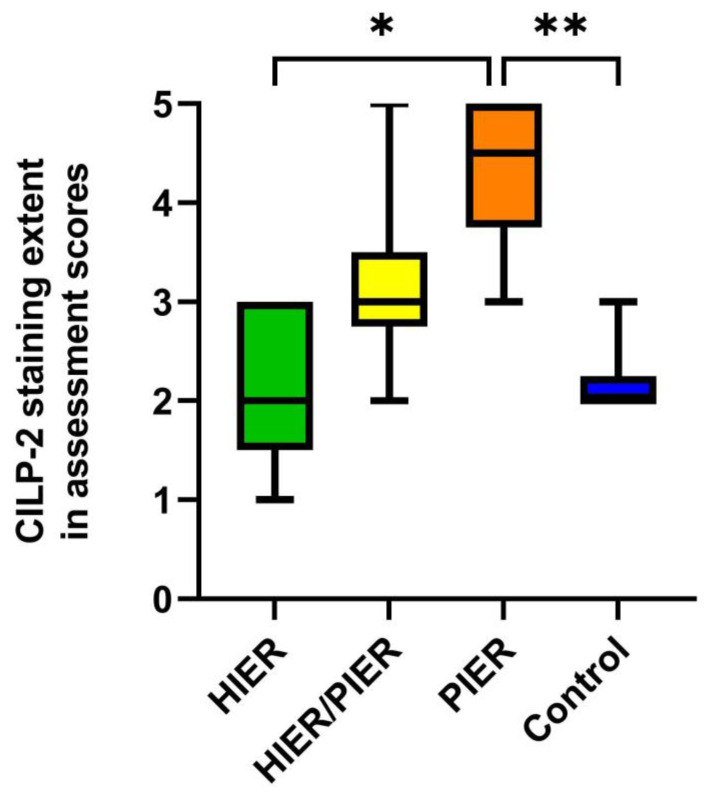
Comparative assessment of CILP-2 IHC staining in four study groups. The groups were formed as follows: HIER group (heat-induced epitope retrieval without proteolytic-induced epitope retrieval), HIER/PIER group (proteolytic-induced epitope retrieval with heat-induced epitope retrieval), PIER group (proteolytic-induced epitope retrieval without heat-induced epitope retrieval), and control group (no antigen retrieval). Grades represent CILP-2 staining extent in assessment scores. Box–whiskers plot with 5th–95th percentiles. Significant differences by the Mann–Whitney U-test: * PIER group vs. HIER group (*p* = 0.012), and ** PIER group vs. control group (*p* = 0.004).

**Figure 3 mps-07-00067-f003:**
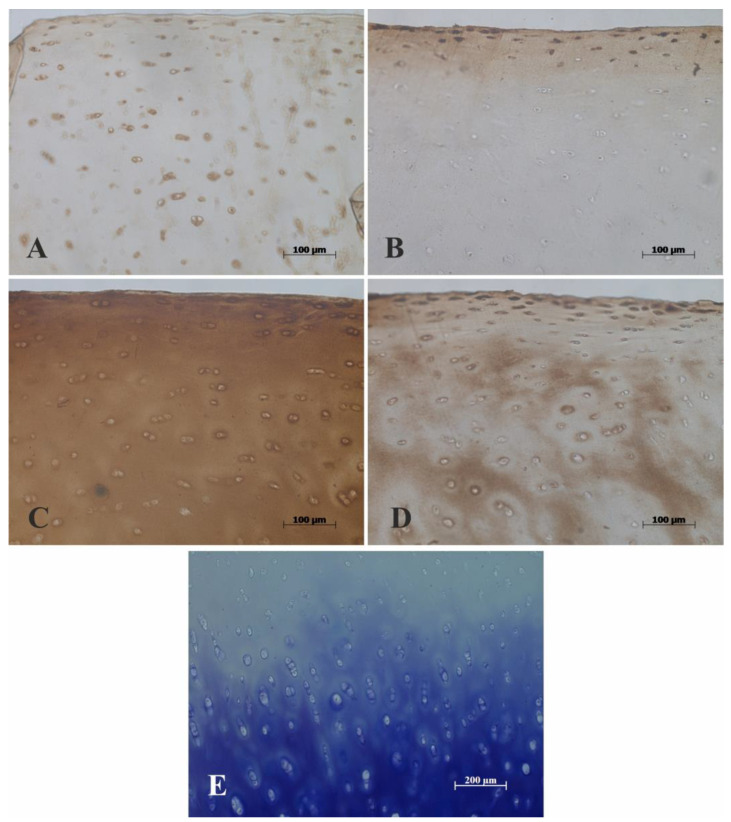
Examples of the effect of different antigen retrieval methods on CILP-2 immunohistochemical staining. (**A**) HIER/PIER—some immunostaining in chondrocytes, imperceptible immunostaining in extracellular matrix. (**B**) HIER—moderate immunostaining in the superficial zone. (**C**) PIER group—abundant immunostaining. (**D**) Control group—moderate immunostaining in the superficial zone and in the middle zone. Toluidine-blue was used for counterstaining. (**E**) Image of negative control where the primary antibody was omitted.

**Figure 4 mps-07-00067-f004:**
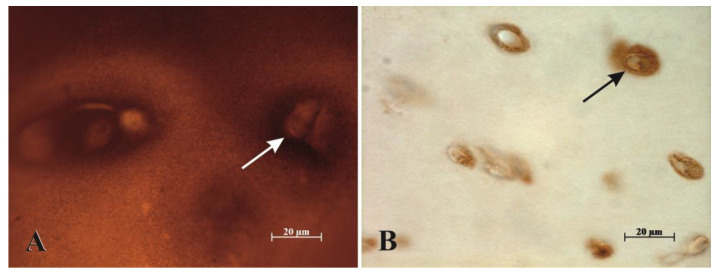
CILP-2 immunohistochemical staining—brown indicates positive staining. (**A**) PIER antigen retrieval. (**B**) HIER/PIER antigen retrieval. CILP-2 staining is noted in the territorial matrix around the isogenous groups of chondrocytes in images (**A**,**B**). The examples are indicated by arrows. The staining of the rest of the tissue matrix is lost in the group that also received heat treatment (image (**B**)), but intensive staining around the isogenous groups remains evident in both images.

## Data Availability

Data are unavailable due to privacy restrictions.

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
