# Peer review of "Comparison of Antigen Retrieval Methods for Immunohistochemical Analysis of Cartilage Matrix Glycoproteins Using Cartilage Intermediate Layer Protein 2 (CILP-2) as an Example"

_mps, 2024, doi:10.3390/mps7050067_

Round 1

Reviewer 1 Report

Comments and Suggestions for Authors

The use of trypsin in immunohistochemistry is the method of choise for antigen retrieval. It was not clear to me the reason that was not evaluated in addition with the use of proteinase K and hyalouronidase

Author Response

Response to Reviewer Comments

Thank you very much for taking time to review this manuscript.

  1. The use of trypsin in immunohistochemistry is the method of choise for antigen retrieval. It was not clear to me the reason that was not evaluated in addition with the use of proteinase K and hyalouronidase.

Response: Thank you for pointing this out. Both trypsin and proteinase K-based methods should be acceptable for epitope retrieval in immunohistochemistry. The choice of enzymatic retrieval method should depend on the specific epitope and tissue under investigation. Enzymatic retrieval using proteinase K has been recommended by other researchers focusing on cartilage (e.g., Juneja et al., 2016; McClellan et al., 2017). The use of hyaluronidase can be considered more specific for articular cartilage, while the application of low-concentration proteinase K is less detrimental to tissue integrity. In contrast, trypsin treatment can cause significant tissue damage (https://ihcworld.com/2024/01/21/trypsin-antigen-retrieval-protocol/), potentially leading to detachment of sections from slides, as described in the manuscript.

Reviewer 2 Report

Comments and Suggestions for Authors

The study investigates four different antigen retrieval protocols: heat-induced epitope retrieval (HIER), proteolytic-induced epitope retrieval (PIER) with proteinase K and hyaluronidase, a combination of HIER and PIER, and a control with no antigen retrieval. The findings that PIER yielded the best CILP-2 IHC staining results are significant. The observation that combining PIER with HIER did not improve and even reduced the staining quality is crucial and provides valuable insights for future IHC protocols.

1.      I would suggest that the authors include staining of additional antigens beyond CILP-2 to validate the effectiveness and generalizability of the antigen retrieval protocols. This would strengthen the study's conclusions.

2.      Figure 2. “Images of negative control samples are presented in the upper right corners of the panels.” However, the image quality is not good enough for publication.

3.      I recommend that the authors include a comparison diagram to summarize and compare the different antigen retrieval methods used in the study.

4.      The manuscript should specify the DAB staining times for each antigen retrieval method.

5.      While the study focuses on OA cartilage, the findings may not be applicable to other types of tissues or conditions.

Author Response

Response to Reviewer 2 Comments

Thank you very much for taking time to review this manuscript.

Comments and Suggestions for Authors: The study investigates four different antigen retrieval protocols: heat-induced epitope retrieval (HIER), proteolytic-induced epitope retrieval (PIER) with proteinase K and hyaluronidase, a combination of HIER and PIER, and a control with no antigen retrieval. The findings that PIER yielded the best CILP-2 IHC staining results are significant. The observation that combining PIER with HIER did not improve and even reduced the staining quality is crucial and provides valuable insights for future IHC protocols.

1. I would suggest that the authors include staining of additional antigens beyond CILP-2 to validate the effectiveness and generalizability of the antigen retrieval protocols. This would strengthen the study's conclusions.

Response: Thank you for pointing this out. While the recommendation to conduct additional experiments with further sets of antigens is logical and greatly valued, it is not feasible to complete these within the 10-day revision period allocated. Given that CILP-2 has the potential to be a distinct research focus in the context of osteoarthritis pathogenesis (as demonstrated by Bernardo et al., 2011 and Boeth et al., 2019), the authors believe it is pertinent to disseminate findings related to the behavior of this specific marker in immunohistochemical experiments. The authors concur that a more comprehensive evaluation involving the primary biomarkers currently central to osteoarthritis research should be pursued. However, they propose that this be the subject of a separate publication, which could reference the present manuscript.

2. Figure 2. “Images of negative control samples are presented in the upper right corners of the panels.” However, the image quality is not good enough for publication.

Response: A better-quality image of the negative control is added to the manuscript (Figure 3-E)

3. I recommend that the authors include a comparison diagram to summarize and compare the different antigen retrieval methods used in the study.

Response: The diagram comparing different retrieval protocols (including reagents and incubation times) is added to the manuscript (Figure 1). 

4. The manuscript should specify the DAB staining times for each antigen retrieval method.

Response: The DAB staining time for each antigen retrieval method is now added.

5. While the study focuses on OA cartilage, the findings may not be applicable to other types of tissues or conditions.

Response: The authors concur with the reviewer's statement, and this has been addressed in the end of discussion section as a partial limitation of the study. However, it is important to note that the immunohistochemistry of cartilage possesses unique characteristics that distinguish it from other tissues. Consequently, we advocate that cartilage immunohistochemistry should frequently be considered independently.